# Analysis of Spatial and Temporal Changes in Vegetation Cover and Driving Forces in the Wuding River Basin, Loess Plateau

Hao Zhang [†] , Zhilin He [†] , Junkui Xu * , Weichen Mu, Yanglong Chen and Guangxia Wang *

College of Geography and Environmental Science, Henan University, Kaifeng 475004, China;
hao6429@henu.edu.cn (H.Z.); hzlgeo@henu.edu.cn (Z.H.); mwc104754210193@henu.edu.cn (W.M.);
chenyanlglong@henu.edu.cn (Y.C.)
* Correspondence: 10130153@vip.henu.edu.cn (J.X.); wangguangxia2011@163.com (G.W.);
  Tel.: +86-15-896-666-097 (J.X.)
[†] These authors contributed equally to this work.

**Abstract:** Vegetation cover in the Loess Plateau region is an important component of ecological protection in the Yellow River Basin, and this study provides a scientific reference for further vegetation restoration. Based on Landsat images and related data, we utilized the dimidiate pixel model and Geodetector method to study the vegetation cover in the Wuding River Basin from 2000 to 2022. The results indicated the spatial and temporal distribution of the vegetation cover and its changes over the study period. Additionally, the driving factors influencing its spatial changes were also uncovered. We also propose a land use shift vegetation cover contribution formula to quantify the effect of land type change on the *FVC*. The study showed that (1) the overall vegetation cover of the watershed increased significantly, and the *FVC* showed an increasing trend from 2000 to 2013 and a slow decline from 2013 to 2022, with the gradual transformation of low-graded *FVC* into a higher graded one. (2) The *FVC* increased spatially from northwest to southeast, and the trend of future changes is mainly decreasing. (3) The strongest explanatory power for the *FVC* change is the land use type and its interactive combination with rainfall. (4) The conversion of grassland to cropland contributes the most to the vegetation cover at 1.52%, and the increase in the cropland area is more conducive to the increase in the vegetation cover.

**Keywords:** contribution of vegetation cover; *FVC*; spatiotemporal analysis; Wuding River Basin; GEE; dimidiate pixel model

## 1. Introduction

The ecological environment serves as the foundation for human survival and development, as well as the basis for economic and social progress. In the past half-century, the intensification of climate change and human activities has posed significant threats to natural ecosystems [1]. As one of the most important components of terrestrial ecosystems, vegetation connects ecological elements such as atmosphere, soil, and hydrology, and plays an important role in maintaining ecosystem stability [2]. Therefore, conducting the dynamic monitoring and trend analysis of vegetation changes across large spatial scales and extended time series has become integral to ecological conservation efforts. Fractional vegetation cover (*FVC*), typically defined as the percentage of land surface area occupied by vertically projected vegetation within an observation area, stands out as one of the most crucial indicators for elucidating ecological changes [3]. In the context of climate change, accurate and long time series of vegetation cover data can better reflect the distribution characteristics and trends of vegetation, and thus provide a scientific basis for ecosystem stability [4]. Based on MODIS-NDVI data, Liu et al. [5] explored the characteristics of spatial and temporal changes in vegetation cover and future trends in the Qinba Mountain region from 2000 to 2014, and analyzed their driving factors. Hao et al. [6] studied the spatial and temporal distribution and change characteristics of vegetation cover in the

Yellow River Basin during 2009–2018 based on SPOT-NDVI data, and predicted its future development trends. Zhang et al. [7] investigated vegetation cover changes and their influencing factors in the Inner Mongolia river section of the Yellow River Basin from 2001 to 2018 using MOD13Q1-NDVI data, and predicted vegetation growth for the next 10 years using the XGBoost method.

Undoubtedly, multi-temporal, high-resolution remote sensing technology provides a rich source of data for vegetation cover studies, and related methods such as physical modeling, empirical modeling, and hybrid image decomposition have been gradually applied [8]. The physical modeling method entails numerous parameters and complex calculations, while the empirical modeling method exhibits limited applicability. Within the hybrid image decomposition method, the dimidiate pixel model posits that a single image comprises two parts: plants and soil. Separate models are constructed for each part to acquire information, resulting in higher accuracy. This approach is primarily used for constructing vegetation cover extraction models [9]. Li et al. [10] employed the dimidiate pixel model to calculate the vegetation cover of the Inner Mongolia Autonomous Region (IMAR) from 2000 to 2013 and conducted an analysis of the trend in the *FVC*. Zhang et al. [11] employed the dimidiate pixel model and the image difference method to process and analyze Landsat data, investigating the spatial and temporal characteristics of vegetation cover in the Santun River Basin. Being one of the classical models in the field of vegetation cover, the dimidiate pixel model is capable of effectively mitigating the effects of factors such as atmospheric or soil background.

The spatial distribution and changes in vegetation cover exhibit unique spatial heterogeneity under the combined influence of intricate natural environments and human activities. Exploring the hidden drivers of regional vegetation cover changes is thus particularly important. Geodetector, a set of statistical methods proposed by Wang [12], can directly quantify the interactions and impacts of driving forces without strictly adhering to traditional statistical assumptions. Zhang et al. [13] explored the drivers of spatial and temporal changes in vegetation NDVI in Inner Mongolia from 2000 to 2015 using the Geodetector method, showing that annual precipitation had the highest explanatory power and dominated the spatial and temporal distribution of vegetation NDVI along with the soil type and vegetation type. Based on MOD13Q1-NDVI data, Li et al. [14] used the Geodetector method to explore the driving factors of spatial and temporal changes in vegetation cover in Ningxia from 2000 to 2020. The results showed the spatial distribution was mainly due to the joint action of three major factors: climate, topography, and human activities. Liu et al. [15] explored the drivers of spatial and temporal variations in NDVI in the Ili River Basin, Xinjiang, from 1998 to 2018 using the Geodetector method. They found that vegetation cover was mainly affected by three factors: temperature, vegetation classification, and altitude, all with explanatory powers over 40%. Many scholars have utilized the Geodetector method to achieve research results, so we also employed it to explore the drivers of vegetation cover change in the Wuding River Basin.

Situated in the middle and upper reaches of the Yellow River in northern Shaanxi, the Loess Plateau region is a typical ecologically fragile area plagued by severe soil erosion. Scholars have shown great concern for the changes in its vegetation cover [16–18]. In 1999, China implemented measures such as "returning farmland to forests and grassland" to combat soil erosion and enhance the ecological environment on the Loess Plateau [19,20]. Yi et al. [21] demonstrated that the NDVI in the growing season of the Loess Plateau showed an increasing trend from 1999 to 2010, and the increase was more significant in the central region. Bai et al. [22] showed that vegetation cover increased significantly in northern Shaanxi, and NDVI was significantly correlated with both precipitation and temperature. Zhang et al. [23] demonstrated that the impact of anthropogenic effects on vegetation cover in the Loess Plateau region is positive. However, the positive effect of anthropogenic effects in the central region is expected to gradually diminish. Previous studies [24,25] have indicated a significant improvement in the vegetation cover of the Loess Plateau over the past decades. However, certain factors in these studies are incomplete. For instance, some

investigations have concentrated solely on the correlation between vegetation cover and climate change [22,26], neglecting the impact of topographic factors and human activities on vegetation growth. While some studies have examined the impact of human activities on vegetation cover [27], there remains a deficiency in quantitative analyses concerning factors like land type change and their effects on vegetation cover.

Based on the GEE remote sensing cloud platform, we extracted the annual average *FVC* of the Wuding River Basin from 2000 to 2022 by the dimidiate pixel model, and applied the Theil–Sen median, Mann–Kendall, and Hurst methods to analyze the characteristics of the changes in the vegetation cover over the past 23 years and to predict trends in the future. We used the Geodetector method to quantify the driving relationships between the *FVC* and meteorological, surface, and anthropogenic factors. In order to explore the impact of human activities on vegetation cover, we propose a formula for the contribution of land use conversion to vegetation cover, quantifying the impact of land transformation on regional vegetation cover. Our findings may provide a theoretical basis for ecological management, soil and water conservation, and contribute to the sustainable development of the Wuding River Basin.

## 2. Materials and Methods

### 2.1. Materials

#### 2.1.1. Study Area

The Wuding River, a primary tributary of the Yellow River, is situated in the Loess Plateau region's heartland (Figure 1), spanning from $107°47'$ E to $110°24'$ E and $37°00'$ N to $39°00'$ N. The basin covers an area of 30,261 km$^2$, exhibiting a topography that is elevated in the southwest and lower in the southeast, characterized by significant undulation with elevations ranging from 597 to 1812 m. The basin experiences a temperate continental monsoon climate with arid and semi-arid conditions. The average annual temperature in the Wuding River Basin ranges from 7.9 °C to 11.2 °C, with an average annual evapotranspiration of approximately 1100–1400 mm. Annual precipitation in the basin is unevenly distributed spatially and temporally, with July-September contributing more than 65% of the annual precipitation. Average precipitation varies from 300 to 500 mm, increasing from northwest to southeast. The main soil types encompass chestnut calcium soil, black clay soil, loess soil, wind sandy soil, freshly accumulated soil, and tidal soil, with loess soil and wind sandy soil being the most widespread. Vegetation is influenced by climate and other environmental factors, displaying a transition from grassland vegetation to desert vegetation from south to north.

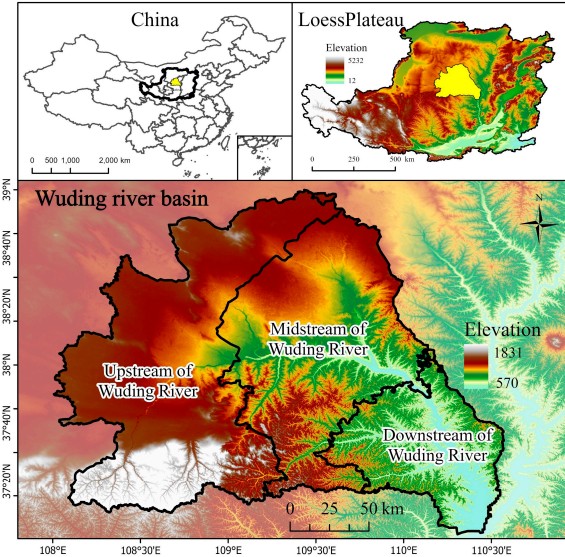

**Figure 1.** Location of the study area.

2.1.2. Data Sources

The study covers the time span of 2000–2022, with the primary data sources being satellite image data, atmospheric data, land data, and anthropogenic data, as presented in Table 1.

**Table 1.** Data sources (All data accessed on 10 September 2023).

| Data Classification | Data | Datasets | Resolution | Data Origins |
|---|---|---|---|---|
| Image | Landsat | Landsat 5/7/8 Surface Reflectance Tier 1 | 30 m | https://earthengine.google.com/ |
| Atmospheric data | Temperature | GPRChinaTemp1 km [28] | 1 km | Zenodo (https://zenodo.org/) |
| | Land surface temperature | China annual land surface temperature dataset | 1 km | Resource and Environmental Science Data Center (www.resdc.cn) |
| | Precipitation | China monthly gridded precipitation [29] | 1 km | Zenodo (https://zenodo.org/) |
| | Relative humidity | China Relative Humidity Dataset | 1 km | National Earth System Science Data Center (www.geodata.cn) |
| | Sunshine hours | China Sunshine Hours Dataset | 1 km | |
| | Aridity index | Global Aridity Index Dataset [30] | 1 km | CGIAR-CSI—Consortium for Spatial Information (wordpress.com) |
| Land data | Soil | Soil Type | 1 km | Resource and Environmental Science Data Center (www.resdc.cn) |
| | Soil erosion | China Soil erosion Dataset | 1 km | |
| | Digital Elevation Mode | NASADEM | 30 m | Earthdata (www.earthdata.nasa.gov) |
| Anthropogenic data | Population density | LandScan Global | 1 km | LandScan (landscan.ornl.gov) |
| | LC | CLCD [31] | 30 m | Zenodo (https://zenodo.org/) |
| | GDP | ChinaGDP [32] | 1 km | |

Note: Except for the image data, the rest of the data were uniformly adjusted to 30 m resolution through resampling.

*2.2. Methods*

The main methodology and research framework used in this study are shown in Figure 2. To examine the specific spatial and temporal changes in vegetation cover within the Wuding River Basin, we segmented the study area into upper, middle, and lower watersheds based on both the river's origin and administrative divisions [33,34]. Based on Landsat imagery, we calculated the interannual *FVC* by selecting the average value from April to October as the interannual NDVI using the maximum value synthesis method and dimidiate pixel model [35]. Meanwhile, the Theil–Sen Median method was employed to analyze the trend of *FVC*, and the Mann–Kendall test was conducted to assess its significance. The coefficient of variation was utilized to analyze the degree of volatility. Ultimately, the Hurst index was applied to predict the development trend of *FVC*. To investigate the driving factors behind spatial and temporal changes in *FVC* in the Wuding River Basin, we selected 13 factors that encompass both natural and anthropogenic elements. Then, we analyzed the dominant factors influencing spatial changes in *FVC* using the Geodetector method and quantified the degree of influence of different factors and their interactions on *FVC*. To explore the impact of anthropogenic factors, particularly land use transformation, on vegetation cover, this paper introduces an innovative formula for the contribution rate of land use transfer to vegetation cover (Equation (10)). This formula is employed to quantify the contribution rate of land use transfer to regional vegetation cover. The details are as follows.

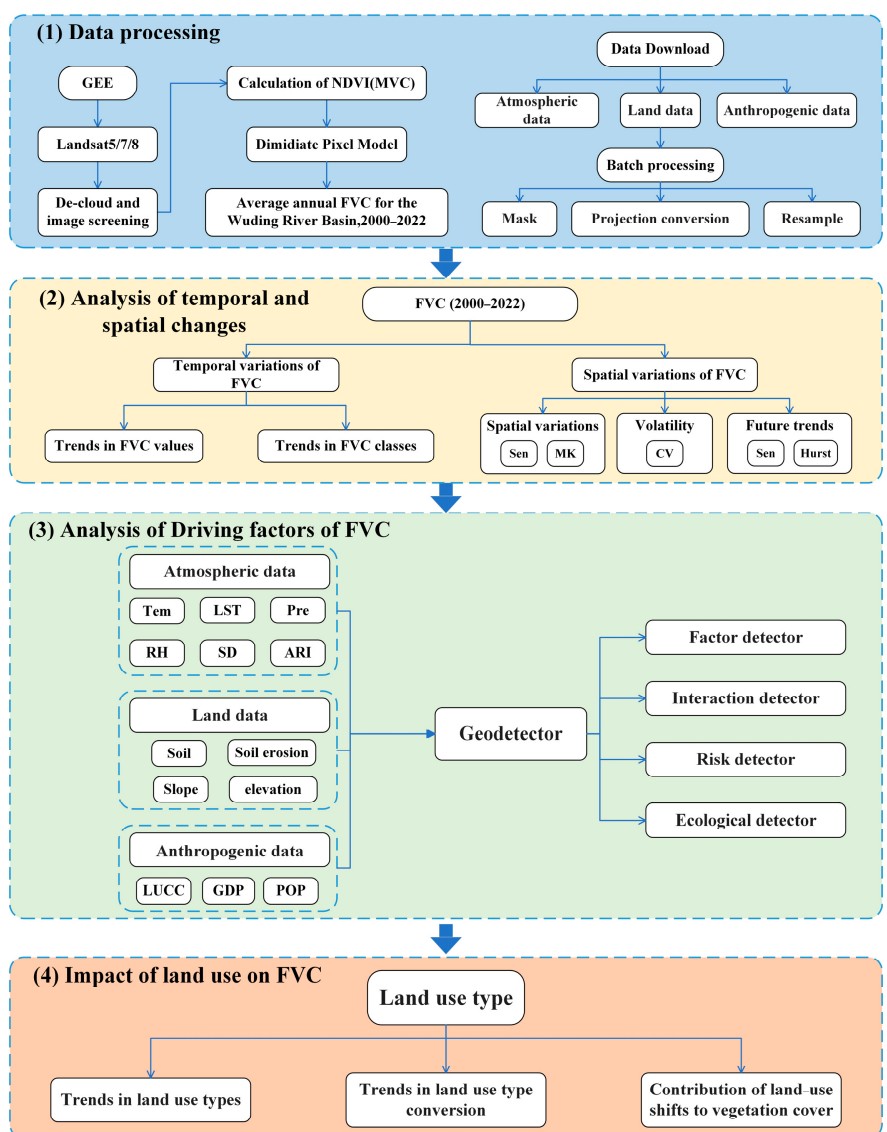

**Figure 2.** Flowchart of technical route.

### 2.2.1. Dimidiate Pixel Model

The dimidiate pixel model assumes that a single image consists of two parts, vegetation and bare soil, and the amount of image information, *S*, consists of the sum of the vegetation part and the soil part, and the *FVC* formula is as follows:

$$FVC = \frac{S - S_{soil}}{S_{veg} - S_{soil}} \tag{1}$$

where *S* is the image information, and $S_{veg}$ and $S_{soil}$, respectively, denote the information reflected by the pure image of vegetation and soil.

We use *NDVI* combined with the dimidiate pixel model to extract the *FVC*:

$$FVC = \frac{NDVI - NDVI_{soil}}{NDVI_{veg} - NDVI_{soil}} \tag{2}$$

In this study, the *NDVI* values at 0.5% and 99.5% of the cumulative frequency were selected to represent $NDVI_{soil}$ and $NDVI_{veg}$, and the *FVC* was classified into five classes using the equally spaced classification [36] (Table 2).

**Table 2.** Criteria for *FVC* classification.

| Class Name | Classification Criteria |
|---|---|
| Lower | $0 < FVC \leq 0.2$ |
| Low | $0.2 < FVC \leq 0.4$ |
| Mediocre | $0.4 < FVC \leq 0.6$ |
| High | $0.6 < FVC \leq 0.8$ |
| Higher | $0.8 < FVC \leq 1.0$ |

2.2.2. Theil–Sen Median and the Mann–Kendall Methods

The Theil–Sen Median, a robust non-parametric method [37,38], is well-suited for trend analysis in long time series data due to its strong error avoidance capability. The formula is as follows:

$$\beta = Median\left(\frac{FVC_j - FVC_i}{j - i}\right), \forall i < j \tag{3}$$

where $\beta$ is the magnitude of the slope, $FVC_j$ and $FVC_i$ are *FVC* data for years $j$ and $i$. *FVC* tends to increase when $\beta > 0$ and decrease when $\beta < 0$.

The Mann–Kendall (MK) test is a non-parametric test for trendiness of time series which does not require the sample to follow a normal distribution and is not disturbed by outliers [39]. When the number of study years is greater than 10, the formulas are as follows:

$$var(S) = \frac{n(n-1)(2n+5) - \sum_{i=1}^{n} t_i(t_i - 1)(2t_i + 5)}{18} \tag{4}$$

$$Z = \begin{cases} \frac{S-1}{\sqrt{var(S)}} & S > 0 \\ 0 & S = 0 \\ \frac{S+1}{\sqrt{var(S)}} & S < 0 \end{cases} \tag{5}$$

where $var(S)$ represents the variance of statistic $S$. We are given a significance level of $\alpha = 0.05$ and when $|Z| > 1.96$, it means that the trend passes the test of significance with a 95% confidence level.

2.2.3. Coefficient of Variation

The coefficient of variation ($C_v$) is expressed as the ratio of the standard deviation to the mean, reflecting the relative volatility of the observed data [40]. The formula is as follows:

$$C_v = \frac{\sqrt{\frac{1}{n}\sum_{i=1}^{n}\left(C_i - \overline{C}\right)^2}}{\overline{C}} \tag{6}$$

The larger the coefficient of variation, the more pronounced the fluctuation of vegetation cover change, and vice versa; the smoother the fluctuation, the more stable the vegetation cover change. By analyzing the coefficient of variation, we can identify specific areas in the study area with significant fluctuations in vegetation cover. These areas may be more severely impacted by human activities and are also key areas for vegetation restoration work.

Referring to [41], $C_v$ was divided into five criteria as shown in Table 3 in order to specifically analyze the degree of fluctuation of *FVC* (Table 3).

**Table 3.** Criteria for $C_v$ classification.

| Class Name | Classification Criteria |
|---|---|
| Lower | $0 < C_v \leq 0.05$ |
| Low | $0.05 < C_v \leq 0.10$ |
| Mediocre | $0.1 < C_v \leq 0.15$ |
| High | $0.15 < C_v \leq 0.20$ |
| Higher | $C_v > 0.20$ |

### 2.2.4. Hurst Index

The Hurst index [40] is commonly used to assess the persistence or inverse persistence in time series trends. For an *FVC* time series $FVC_{(\tau)}$, with $\tau$ = 1, 2, 3, 4, ..., *n*, its mean value is

$$\overline{FVC}_{(\tau)} = \frac{1}{\tau}\sum_{t=1}^{\tau} FVC_{(\tau)} \qquad \tau = 1, 2, 3, \ldots, n \tag{7}$$

The sequence of cumulative deviations $X_{(t,\tau)}$ is

$$X_{(t,\tau)} = \sum_{j=1}^{t}\left(FVC_{(t)} - \overline{FVC}_{(\tau)}\right) 1 \leq t \leq \tau \tag{8}$$

The extreme difference is

$$R_{(\tau)} = \max_{1 \leq t \leq \tau} X_{(t,\tau)} - \min_{1 \leq t \leq \tau} X_{(t,\tau)} \tag{9}$$

The standard deviation is

$$S_{(\tau)} = \sqrt{\frac{1}{\tau}\sum_{t=1}^{\tau}\left(FVC_{(t)} - \overline{FVC}_{(\tau)}\right)^2} \tag{10}$$

The Hurst index is as follows:

$$\frac{R_{(\tau)}}{S_{(\tau)}} = (c\tau)^H \tag{11}$$

where *H* is the Hurst index, the effective value range is 0–1, and its meaning [42] is shown in Table 4.

**Table 4.** Criteria for Hurst index classification.

| Type of Change | Classification Criteria |
|---|---|
| Anti-persistence | 0 < Hurst < 0.5 |
| Random | Hurst = 0.5 |
| Persistence | 0.5 < Hurst < 1.0 |

### 2.2.5. Geodetector

Geodetector is a set of statistical methods for detecting spatial dissimilarity and revealing driving forces behind it (Table 5). The core idea is based on the assumption that if an independent variable significantly affects a dependent variable, then the spatial distributions of the independent and dependent variables should be similar [12]. We selected a total of 13 drivers from a combination of meteorological, surface, and anthropogenic factors, all using the latest 2022 data (Table 6). We used the 2022 *FVC*, processed by the maximum value synthesis method and the dimidiate pixel model, as the dependent variable Y. We input the 13 driving factors as the independent variable X into the geoprobe to explore the response relationship between vegetation cover changes and factors in the Wuding River Basin.

**Table 5.** Role of the different modules of the Geodetector.

| Type of Detector | Function of Detector |
|---|---|
| Factor detector | Detecting the extent to which a factor X explains the spatial dissimilarity of an attribute Y. |
| Interaction detector | Identify interactions between different influences on factor X, i.e., assess whether factors X1 and X2, acting together, increase or decrease the explanatory power of the dependent variable Y. |
| Risk detector | Determine whether the mean values of the attributes of the corresponding independent variable Y are significantly different in different intervals of the X factor. |
| Ecological detector | Compare whether there is a significant difference between the effects of the two factors X1 and X2 on the spatial distribution of attribute Y. |

**Table 6.** Factors applied to Geodetector.

| Data Classification | Factor | Unit |
|---|---|---|
| Atmospheric factor | Temperature | °C |
| | Land surface temperature | °C |
| | Precipitation | mm |
| | Relative humidity | % |
| | Sunshine hours | h |
| | Aridity index | - |
| Land factor | Soil | - |
| | Soil erosion | - |
| | Slope | ° |
| | DEM | m |
| Anthropogenic factor | Population density | persons/km$^2$ |
| | LC | - |
| | GDP | million CNY |

2.2.6. Contribution of Land Use Shifts to Vegetation Cover

Human activities, mainly land use types, have a significant impact on vegetation cover, and land use change driven by relevant policies is one of the main drivers of vegetation cover change in the Loess Plateau region [27]. However, there is still a lack of research to quantitatively analyze the impact of land use transformation on regional vegetation cover; in order to solve this problem, this paper innovatively proposes a formula for the contribution rate of land use transfer vegetation cover, as shown in Equation (12):

$$FVCR = (FVC_1 - FVC_0) \times \frac{LA}{TA} \tag{12}$$

where $FVCR$ is the contribution rate of vegetation cover, $FVC_1$ and $FVC_0$ represent the average vegetation cover at the end and beginning of the land use change period. $LA$ is the land area of the change type; $TA$ is the total area of the study area.

The formula enables the direct quantification of the contribution of land use transformation to vegetation cover. This facilitates the analysis of which type of land use change has the most significant impact on regional vegetation cover, offering valuable insights for informed suggestions on regional vegetation restoration and ecological protection.

## 3. Results

*3.1. Characteristics of Temporal Changes in FVC*

3.1.1. Temporal Trends in *FVC*

The analysis of the time variation in the *FVC* in the study area from 2000 to 2022 by linear regression as well as a *t*-test shows that (Figure 3) the mean value of the *FVC* in the study area during the 23-year period was 0.34, with an overall upward trend, and the average annual growth rate S was 0.0046/a ($p < 0.01$). The average annual *FVC* increases from 0.26 in 2000 to 0.41 in 2022, representing a 57.7% increase. The lowest and highest values are observed in 2000 (0.26) and 2013 (0.43). During the period of 2000–2013, the average annual *FVC* in the Wuding River Basin exhibited a rapid growth trend, with a growth rate S of 0.011/a ($p < 0.01$). The mean value of the *FVC* over 14 years was 0.33, with the mean value in 2013 reaching 0.43, representing a 62.7% increase compared to 2000. The *FVC* increased significantly in all three watershed segments of the Wuding River Basin, with the highest rate of *FVC* growth in the lower reaches, at a rate S of 0.0172/a ($p < 0.01$); the slowest rate of *FVC* growth in the upper reaches, at a rate S of 0.0076/a ($p < 0.01$); and the mid-reaches, at a rate S of 0.0106/a ($p < 0.01$).

During the period 2013–2022, the *FVC* exhibited a fluctuating downward trend, with an annual average decline rate S of $-0.0027/a$ ($p < 0.01$), and the 10-year average *FVC* value of 0.37. All the watershed segments showed a declining trend, with the downstream area experiencing the most pronounced decline, having a decline rate S of $-0.0077/a$ ($p < 0.01$),

the midstream area showing a decline rate of −0.0017/a ($p < 0.01$), and the upstream area demonstrating the slowest decline, with a decline rate S of −0.0009/a ($p < 0.01$). Overall, the *FVC* in the Wuding River Basin was in a period of rapid growth during 2000–2013, and after reaching a peak growth rate in 2013, it began to fluctuate and decline, while in 2021, it showed a recovery of vegetation. The average *FVC* in the three watershed segments were 0.45 in the lower reaches, 0.33 in the middle reaches, and 0.31 in the upper reaches. The downstream area experienced the fastest growth and decline in *FVC*, while the upstream area had the slowest growth and decline. However, the middle reaches exhibited an overall better growth rate compared to the downstream area.

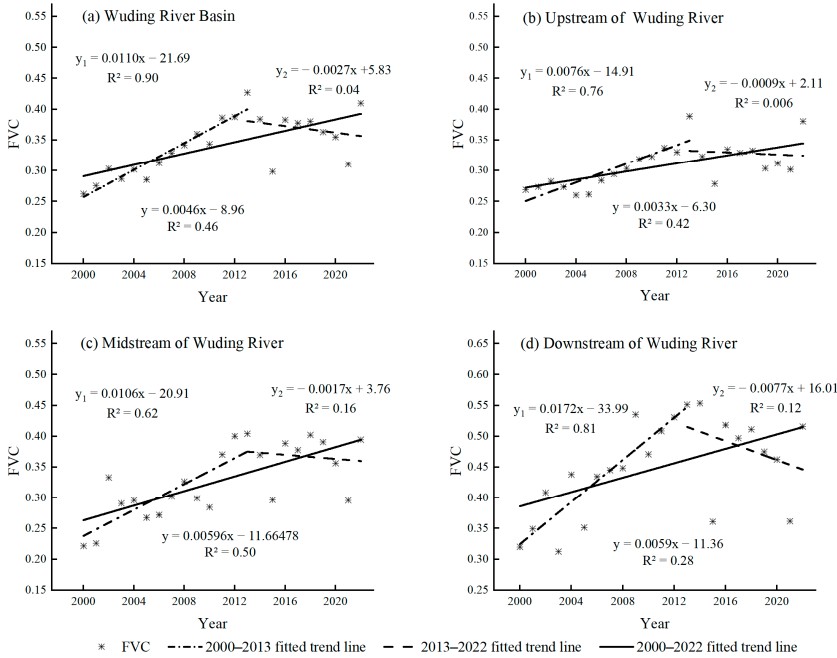

**Figure 3.** Temporal trends of *FVC* in the Wuding River Basin and different basin segments.

### 3.1.2. Characteristics of *FVC* Class Changes

Based on Table 2, the *FVC* for the Indeterminate River Basin from 2000 to 2022 is categorized into different classes (Figure 4). Throughout the time series, the proportion of Lower area in the study region decreased significantly, while the percentage of Low area exhibited an initial increase followed by a subsequent decline. Meanwhile, the percentages of Mediocre, High, and Higher area experienced significant increases. Characteristic changes in the vegetation cover classes varied considerably between the pre- and post-2013 phases. From 2000 to 2013, the proportion of Lower area decreased significantly, reaching a minimum of 13% in 2013. During this period, the proportion of Low area increased and then decreased, reaching a minimum of 33% in 2013. Moreover, the proportions of Mediocre and High area steadily increased, reaching a maximum of 35% and 14%, respectively, in 2013, while the proportion of Higher area increased to a lesser extent.

From 2013 to 2022, the proportion of Lower and Low areas exhibited an increase followed by a decrease, while the proportion of Mediocre and High areas showed a decrease followed by an increase. The proportion of Higher areas fluctuated, reaching a maximum of 5% in 2022. This suggests a certain degree of vegetation degradation in the study area during this period, but is gradually improving [43]. Analyzing the overall changes over the 23-year period reveals a 29% decrease in the proportion of Lower area, a 4% decrease in the proportion of Low area, a 24% increase in the proportion of Mediocre area, a 7% increase in the proportion of High area, and a 3% increase in the proportion of Higher area. Overall, the increasing percentage of Mediocre and High area is primarily due to improved vegetation in the Lower and Low classes. The vegetation cover in the Wuding River Basin is improving, indicating the effectiveness of vegetation restoration [16].

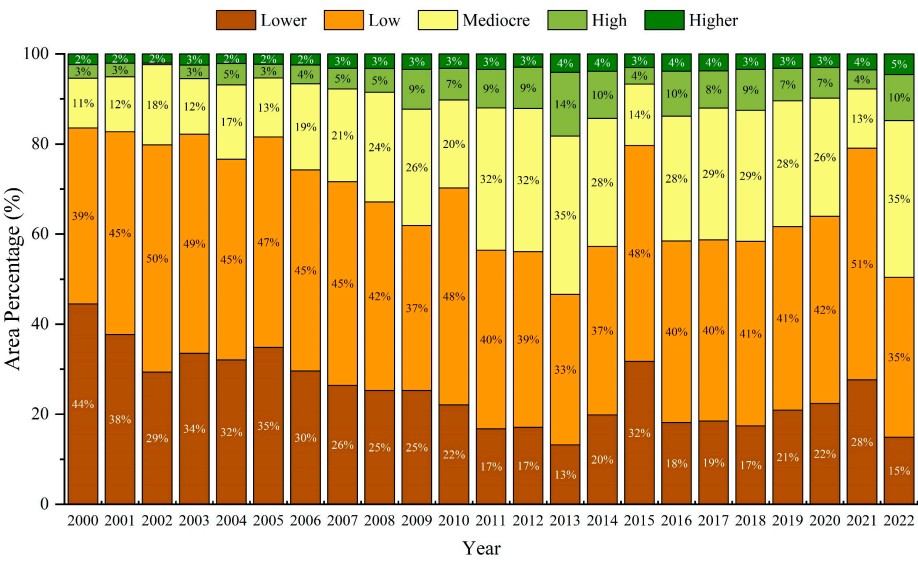

**Figure 4.** Area percentage of different classes of *FVC* in the Wuding River Basin.

Statistics on the transfer of the different grades of *FVC* from 2000 to 2022 are shown in Table 7. In 2022, the area of Lower and Low decreased by 8950.95 km$^2$ and 4941.43 km$^2$, respectively. Meanwhile, the area of Mediocre, High, and Higher areas increased by 5360.17 km$^2$, 1871.98 km$^2$, and 343.98 km$^2$, respectively. The most substantial decrease occurred in the area share of the Lower class, transferring out of this type by 9707.33 km$^2$, followed by the Relatively Lower class with 7939.32 km$^2$. The Lower class predominantly converted into Relatively Low and Mediocre, while Low transformed into High and Higher. Overall, the *FVC* classes underwent significant optimization during 2000–2022, resulting in a significant improvement in vegetation cover.

**Table 7.** *FVC* Class Transfer Matrix (km$^2$).

| 2000 | 2022 | | | | | | |
|---|---|---|---|---|---|---|---|
| | **Lower** | **Low** | **Mediocre** | **High** | **Higher** | **Total** | **Roll-Out** |
| Lower | 3756.43 | 6031.05 | 2980.51 | 426.45 | 269.33 | 13,463.76 | 9707.33 |
| Low | 599.02 | 3869.46 | 5650.22 | 1354.38 | 335.70 | 11,808.78 | 7939.32 |
| Mediocre | 89.97 | 659.87 | 1560.47 | 782.02 | 253.74 | 3346.07 | 1785.60 |
| High | 28.69 | 107.89 | 249.38 | 310.10 | 215.77 | 911.83 | 601.73 |
| Higher | 38.70 | 68.54 | 96.14 | 220.96 | 306.21 | 730.55 | 424.34 |
| Total | 4512.82 | 10,736.81 | 10,536.72 | 3093.91 | 1380.75 | | |
| Roll-in | 756.39 | 6867.35 | 8976.24 | 2783.81 | 1074.53 | | |
| Variation | −8950.95 | −4941.43 | 5360.17 | 1871.98 | 343.98 | | |

Note: (Roll-in is the transformation of other classes into this class, roll-out is the transformation of this class into other classes and variation is the amount of change in a period minus the initial period).

### 3.2. Characteristics of Spatial Changes in FVC

#### 3.2.1. Characteristics of *FVC* Spatial Distribution

As shown in Figure 5, the *FVC* shows an increasing spatial distribution from north to south and from west to east from 2000 to 2022, with significant geographical differentiation. The *FVC* low value area is mainly located in Wushen Qi as well as in the northwestern part of Yuyang Qu. This region is also known as the Mao Wusu Desert, characterized by poor vegetation cover. However, high-grade *FVC* also exists, suggesting some advancements in ongoing afforestation activities [44]. The high *FVC* area is primarily situated in the southeastern part of the study area, encompassing the middle and lower sections of the watershed. This region constitutes a loess hill and gully area, mainly comprising Mili County, Suide County, and Zizhou County, characterized by windy sandy beaches, flat terrain, abundant groundwater, and favorable conditions for vegetation growth.

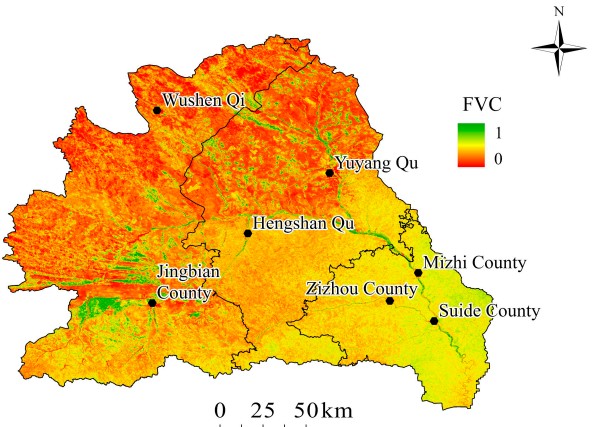

**Figure 5.** Spatial distribution pattern of mean vegetation cover in the Wuding River Basin, 2000–2022.

To further investigate the distribution of vegetation, we classified the average *FVC* according to Table 2 (Figure 6). The dominant *FVC* type was Low (48.7%), followed by Mediocre (26.4%), with area shares of Lower, High, and Higher at 19.1%, 3.9%, and 1.8%. Analyzing the area distribution of the *FVC* classes in the upper, middle, and lower sections of the Wuding River Basin reveals that in the upper and middle reaches, Low has the largest area share, followed by Low. In the lower reaches, Mediocre has the largest area share, accounting for 70% of the total area share. This is because most of the upper and middle reaches of the Wuding River basin are in the Maowusu Desert, characterized by sparse vegetation, primarily desert vegetation. In contrast, the downstream area belongs to the loess hill and gully area, more conducive to vegetation growth than the rest of the basin, featuring predominantly short plants with a sparse and simple population composition [45].

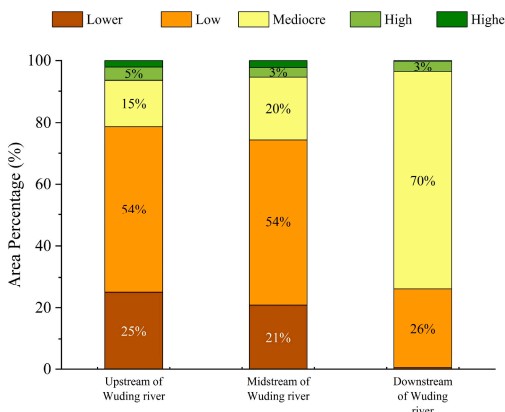

**Figure 6.** Percentage of area in *FVC* class in three watershed segments.

### 3.2.2. Characteristics of *FVC* Spatial Variation

We employed Theil–Sen Median and Mann–Kendall trend analysis to investigate the spatial and temporal changes in the *FVC* in the Wuding River Basin from 2000 to 2022 (Figure 7). In 45.9% of the areas in the Wuding River Basin, the *FVC* exhibited an increasing trend, with 12.6% of the areas showing a significant increase and 33.3% showing a slower increase. The areas with a significant increase were primarily in the south of the middle reaches of the Wuding River and the lower reaches, suggesting the highly effective implementation of the project to convert farmland back into forests and grassland [46]. About 51.5% of the areas in the Wuding River Basin experienced a decreasing trend in the *FVC*, with 48% of them exhibiting a slower decline, while 3.5% showed a significant decline. The areas with significant decline were mainly located in the northwest of the study area, namely, Wushen Qi and Yuyang District, situated at the southern edge of the Mao Wusu Desert, indicating certain vegetation degradation in the desert region [21].

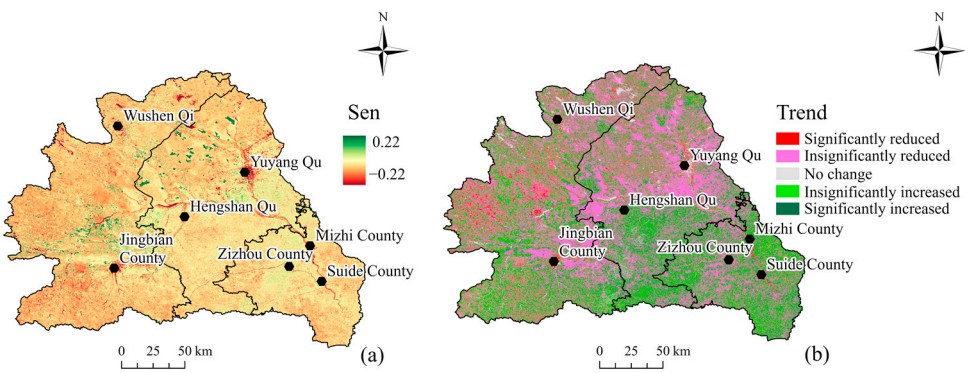

**Figure 7.** Characteristics of spatial variation in *FVC* in the Wuding River Basin, 2000–2022. (**a**) Theil–Sen Median trend analysis; (**b**) Mann–Kendall significance test.

The fluctuation of *FVC* is illustrated in Figure 8. Overall, the average value of the coefficient of variation (CV) for the 23-year period is 0.29, indicating a high degree of fluctuation. Among the five variability levels, Mediocre comprises the largest share, amounting to 35.1%. This is mainly observed in counties such as Mili and Zizhou in the lower basin section of the Wuding River. Low and Lower volatility zones contribute to 8.5% of the total, with a more scattered distribution, mainly in Jingbian County within the upper basin of the Indeterminate River and the northern end of the middle reaches of Yuyang District, among other places with large-scale farmland. The High and Higher-volatility zones, making up 56.4% of the total, are primarily situated in the wind-sand area of the upper basin of the Wuding River in the northwest of the study area. This area has a harsh natural environment and is heavily influenced by ecological management and other factors, resulting in large fluctuations in the regional vegetation cover. The High and Higher zones, accounting for 56.4%, are mainly located in the sandy and windswept area northwest of the upper reaches of the Wuding River, where the natural environment is harsh and is directly affected by anthropogenic factors, such as desert modification, thus contributing to a high degree of volatility in the regional vegetation cover.

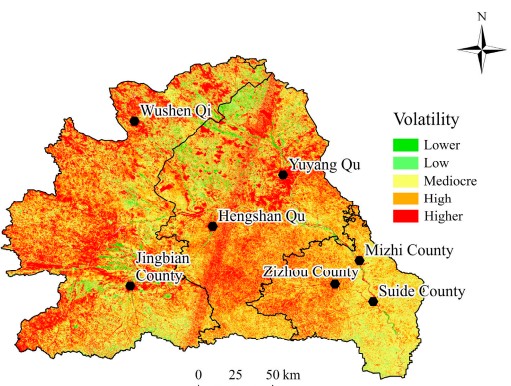

**Figure 8.** Spatial distribution pattern of the degree of volatility of vegetation cover change in the Wuding River Basin, 2000–2022.

The Hurst index of the *FVC* is illustrated in Figure 9a, which has a maximum value of 0.99. The majority of the areas (64.6%) exhibit an anti-continuous future trend, primarily located in the southeast of the watershed, particularly the downstream areas of Mili County and Suide County, which are more influenced by human activities. In the watershed, 35.3% of the area shows persistence, concentrated in the northwest of the study area, namely the upper basin of the Wuding River and the northern end of the middle reaches of the river, among other areas with large-scale farmland. These areas are less volatile and also exhibit some persistent changes, as indicated by the analysis of *FVC* volatility.

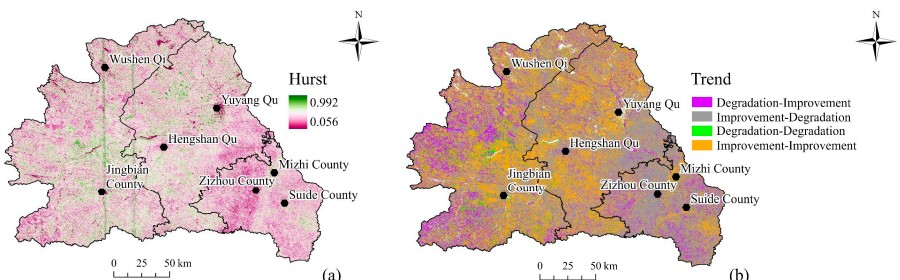

**Figure 9.** Predicting future trends in *FVC*. (**a**) Hurst index analysis; (**b**) future trends in *FVC*.

In this study, Hurst and Sen were superimposed to further explore the future trends of vegetation change. The superposition of three Hurst and two Sen values produces six results. Ignoring Hurst values of 0.5 as they account for a small percentage and are difficult to predict, we focus on the remaining four: anti-sustained growth, anti-sustained decline, sustained growth, and sustained decline (Figure 9b). The future trend in the *FVC* is dominated by decline, with a regional share of 58.0%. The largest proportion of anti-persistent decline is 50.1%, concentrated in Zizhou County and Mili County in the downstream region of the Wuding River, likely due to urbanization and development affecting regional vegetation cover [47]. And the proportion of persistent decline (degradation—degradation) was 8.0%, mainly distributed in the Wuxing Banner and Jingbian County regions in the upper part of the Wuding River region. The proportion of sustained growth (improvement—improvement) was 27.3%, which was mainly distributed in Hengshan District in the middle reaches of the Wuding River Basin region as well as in the northern end of Jingbian County. The proportion of anti-sustained growth is 14.6%, distributed similarly to (degradation—degradation), indicating afforestation and other work in the Upper Wuding River's upper part needs improvement, and some degraded areas remain untransformed. Overall, there is a trend of improvement followed by degradation in the upper reaches of the Wuding River, while the improvement of vegetation cover in the middle and lower reaches of the river has improved, but there is still a need to strengthen the restoration of vegetation cover.

### 3.3. Analysis of Factors Driving Changes in FVC

#### 3.3.1. Factor Detector

The spatial and temporal analyses of the vegetation cover revealed pronounced spatial heterogeneity in its distribution. The analysis of the factors influencing the spatial variability of the *FVC* using the factor detector (Table 8) demonstrated that while all the factors significantly influenced the *FVC*'s spatial variability ($p < 0.01$), the explanatory power (q) varied widely. The q value is the explanatory power of the driver on the spatial distribution of the *FVC*, with larger q values indicating stronger influence and smaller q values weaker influence. q has a value range of [0, 1], with a value of 1 indicating complete control of the spatial distribution of the *FVC* by the driver, and a value of 0 indicating no relationship between the driver and *FVC*.

**Table 8.** Explanatory power and significance of driving factors.

| Factor | Tem | LST | Pre | RH | SD | Aridity Index | Soil | Soil Erosion | Elevation | Slope | Pop | LC | GDP |
|--------|------|------|------|------|------|------|------|------|------|------|------|------|------|
| q | 0.064 | 0.111 | 0.133 | 0.097 | 0.123 | 0.121 | 0.117 | 0.056 | 0.088 | 0.038 | 0.108 | 0.345 | 0.011 |
| p | 0.000 | 0.000 | 0.000 | 0.000 | 0.000 | 0.000 | 0.000 | 0.000 | 0.000 | 0.000 | 0.000 | 0.000 | 0.000 |

Note: Tem, air temperature; LST, land surface temperature; Pre, precipitation, RH, relative humidity; SD, sunshine duration; Pop, population density; LC, land cover; GDP, gross domestic product. q is the explanatory power of each factor in the Geodetector's factor detection for the spatial distribution of *FVC* and p is the level of statistical significance.

The explanatory power, in decreasing order, included land cover, precipitation, sunshine hours, aridity index, soil type, population density, surface temperature, relative humidity, altitude, temperature, soil erosion, slope, and GDP. Among them, the q-values for anthropogenic factors, land use type, and meteorological factors (precipitation, sunshine hours, and aridity index) were greater than 12%, specifically 34.5%, 13.3%, 12.3%, and 12.1%, respectively. This suggests that the spatial variation in vegetation cover in the Wuding River Basin results from a combination of human activities and meteorological factors, with land use patterns playing a significant role in determining spatial variability [20]. Soil type, population density, surface temperature, relative humidity, altitude, temperature, and soil erosion had a relatively weak explanatory power for spatial variation in *FVC*, with the q-values for slope and GDP below 5%, indicating low influence.

### 3.3.2. Interaction and Ecological Detector

We assessed the explanatory power of the changes in the *FVC* after factor interactions based on an interaction detector (Figure 10); the results revealed a significant interaction effect among the factors influencing the *FVC*. The effects of these factors were not independent but occurred synergistically. The interaction effects included two-factor enhancement and nonlinear enhancement, with the latter being more pronounced than the former. The most significant explanatory power in the interaction is associated with land cover and precipitation, with a q-value reaching 0.523. The interaction between GDP and slope exhibits the lowest explanatory power, with a q-value of only 0.055. This may be due to the limited explanatory ability of both the slope and GDP individually in relation to the *FVC*, leading to insufficient explanatory power when combined. Simultaneously, the combination of land cover with most factors can yield a high q-value and its combination with precipitation, sunshine hours, and soil type is optimal, with q-values exceeding 0.5.

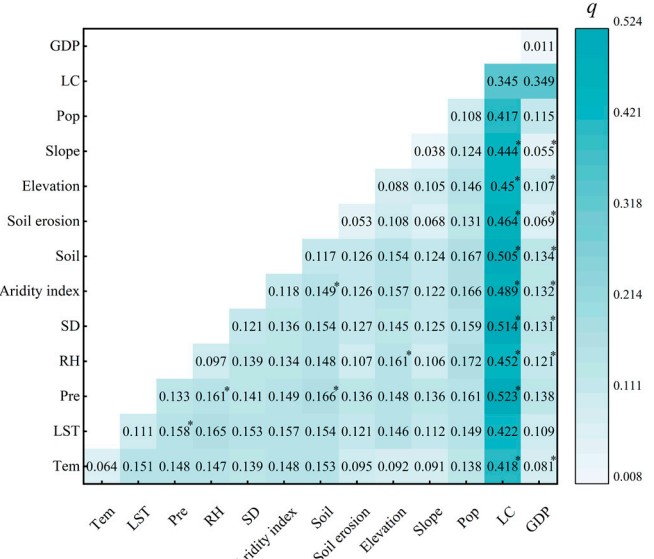

**Figure 10.** Interaction detector (non-linear enhancements marked with * in the figure, the rest are two-factor enhancements).

Employing the ecological detector, we conducted a comparative analysis to determine the significant differences between each pair of factors regarding the spatial distribution of the *FVC*. As depicted in Table 9, there are notable distinctions in the impact of most factors on the spatial distribution of the *FVC* when comparing each pair of factors. However, the population density does not exhibit significant differences when compared to most factors.

**Table 9.** Ecological detector results.

| Factor | Tem | LST | Pre | RH | SD | Aridity Index | Soil | Soil Erosion | Elevation | Slope | Pop | LC | GDP |
|---|---|---|---|---|---|---|---|---|---|---|---|---|---|
| Tem | | | | | | | | | | | | | |
| LST | Y | | | | | | | | | | | | |
| Pre | Y | Y | | | | | | | | | | | |
| RH | Y | N | Y | | | | | | | | | | |
| SD | Y | Y | N | Y | | | | | | | | | |
| Aridity index | Y | Y | Y | Y | N | | | | | | | | |
| Soil | Y | Y | Y | Y | N | N | | | | | | | |
| Soil erosion | N | Y | Y | Y | Y | Y | Y | | | | | | |
| Elevation | Y | N | Y | N | Y | Y | Y | Y | | | | | |
| Slope | Y | Y | Y | Y | Y | Y | Y | N | Y | | | | |
| Pop | Y | N | Y | N | N | N | N | Y | Y | Y | | | |
| LC | Y | Y | Y | Y | Y | Y | Y | Y | Y | Y | Y | | |
| GDP | Y | Y | Y | Y | Y | Y | Y | Y | Y | Y | Y | Y | |

Note: Y represents a significant difference between the two factors for the spatial distribution of *FVC*, while N represents no significant difference.

### 3.3.3. Risk Detector

Based on the risk detector, we determined the mean *FVC* values across various ranges of factors or types of vegetation growth (Figure 11). Concerning meteorological factors, the *FVC* exhibited a pattern of initially decreasing and then gradually increasing with rising temperature, reaching the maximum value at temperatures of 18–20 °C. Similarly, with increased precipitation, the *FVC* showed an increasing trend, peaking at precipitation levels of 601–648 mm. The *FVC* showed an initial rise before declining, peaking at 53.4%–54.4% relative humidity. It also rose then fell with increasing sunshine, peaking at 7.86–8.07 h. Vegetation thrives in the meteorological environment of high temperature, low sunshine, high rainfall, and medium humidity in the Wuding River Basin.

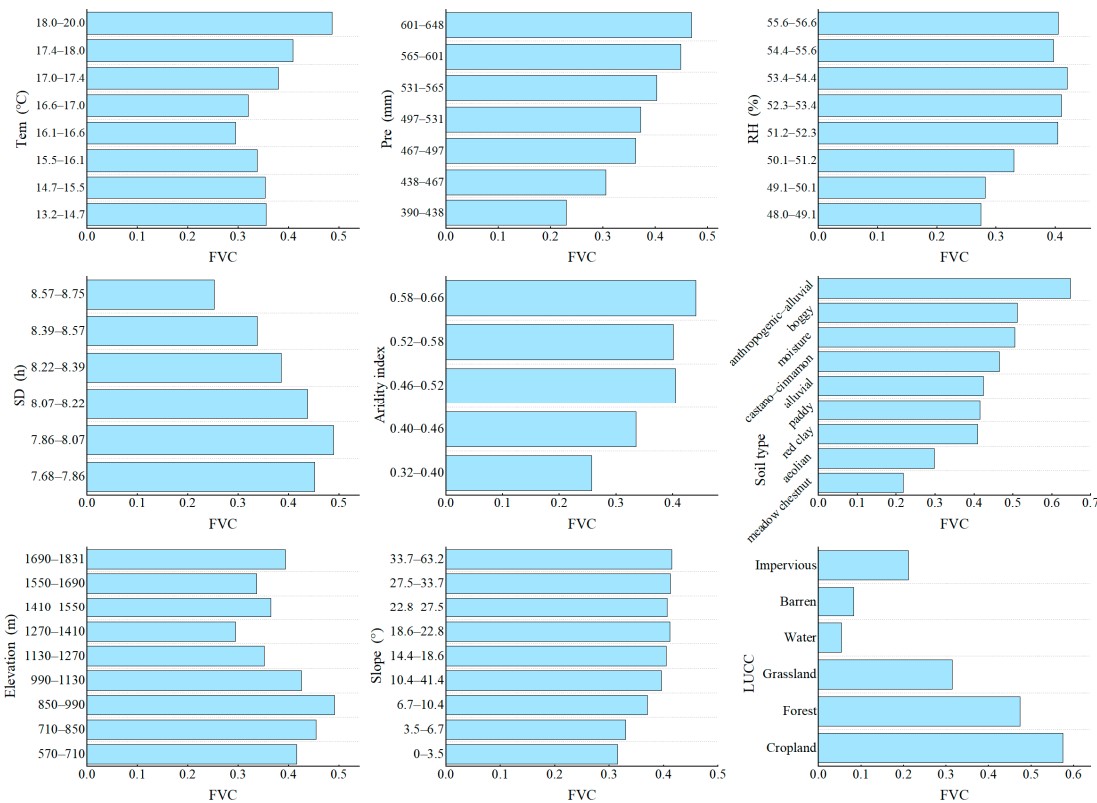

**Figure 11.** Risk detector results.

Concerning surface factors, *FVC* exhibited a decreasing trend with an increase in the aridity index, reaching its maximum at an aridity index of 0.58–0.66 (lower aridity index indicates higher aridity) [30]. Changes in the soil type correspondingly influenced the *FVC* in the Wuding River Basin, with the largest vegetation cover observed in irrigated silt soil, signifying its suitability for vegetation growth in the basin. Additionally, as the elevation increased, *FVC* initially rose, then declined, and eventually stabilized, peaking at elevations between 850 and 990 m. Similarly, with an increase in slope, the *FVC* demonstrated an upward trend, reaching its maximum value at slopes between 16.8 and 22.8°. Notably, the soil type factor exhibited the highest q-value and the most robust explanatory power, highlighting its pivotal role as a surface driver of vegetation change in the Wuding River Basin. Regarding anthropogenic factors, the Land Cover (LC) possessed the highest q-value among all the factors, indicating that a high *FVC* could be achieved when the land use types in the study area were farmland and grassland.

### 3.4. Impact of Land Use Type Shifts on Vegetation Cover

#### 3.4.1. Trends in Land Use Types

The predominant land use types in the Wuding River Basin, in descending order, are grassland, cropland, barren, impervious, water, and forest. Over the 23-year period, they account for an average share of 73.58%, 20.30%, 5.52%, 0.24%, and 0.07%, respectively, with grassland being the dominant component, followed by cropland. Illustrated in Figure 12, grassland, impervious, water, and forest exhibit noteworthy growth trends, while cropland and barren manifest a distinct declining trend. And grassland increased the most, with an increase rate S of 0.00548/a ($p < 0.01$), while cropland decreased the most rapidly, with a decrease rate S of 0.00138/a ($p < 0.01$). Overall, the increase in grassland, water bodies, and woodland indicated that the overall *FVC* of the Wuding River Basin showed an increasing trend. And the decrease in cropland and wasteland, along with the increase in impervious surface, indicates that the urban development process in the Wuding River Basin is also progressing gradually [48].

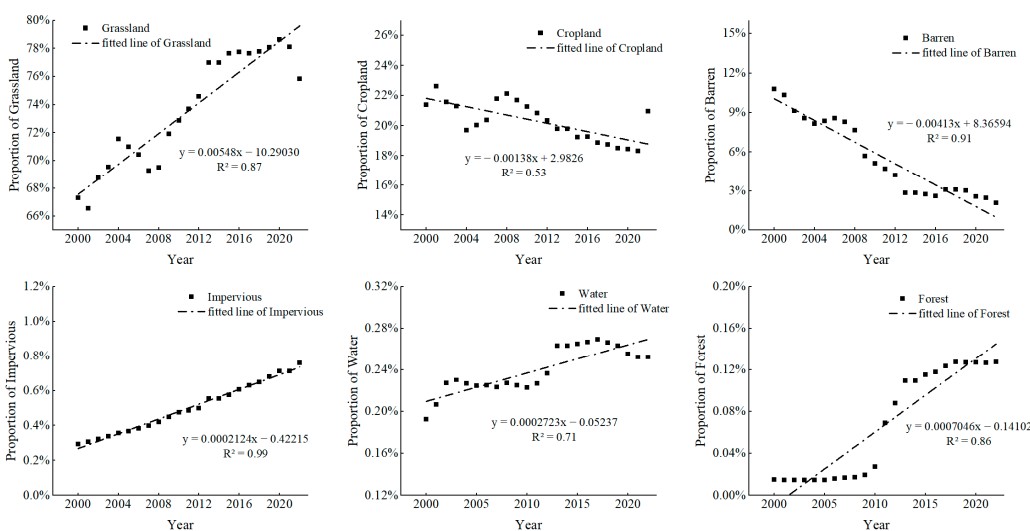

**Figure 12.** Changes in land use types in the Wuding River Basin, 2000–2022.

#### 3.4.2. Land Use Type Shifts and Contribution of Vegetation Cover

In the previous study, the *FVC* of the Wuding River Basin experienced rapid growth from 2000 to 2013. After reaching its peak growth rate in 2013, it began to fluctuate and decline from 2013 to 2022. The characteristics of the *FVC* changes varied significantly between the two periods. In 2013, compared to 2000, grassland increased by 2799.23 km$^2$, barren decreased by 2403.58 km$^2$, cropland decreased by 522.83 km$^2$, impervious increased by 77.67 km$^2$, forest and water slightly increased. These changes suggest the successful

implementation of the policy to convert farmland back to forest and grassland in the Wuding River Basin during this period [49]. Figure 13 illustrates that the majority of the decreased wasteland and arable land underwent conversion to grassland, with the exception of maintaining itself unchanged. This observation supports the notion that the *FVC* in the Wuding River Basin experienced rapid growth between 2000 and 2013. In contrast, from 2013 to 2022, there was a reduction in grassland by 215.41 km$^2$, a decrease in wasteland by 236.88 km$^2$, whereas cropland increased by 386.4 km$^2$. The growth of forest land was notably lower compared to the 2000–2013 period, and there was a decrease in water area. These changes suggest that the accelerated urbanization during this period had a more pronounced impact on vegetation cover, leading to degradation in some areas [43]. The conversion of grassland to cropland and barren increased significantly compared to the pre-2013 period, attributed partly to shifts in national policies regarding arable land resources [46], and partly to environmental deterioration from human activities.

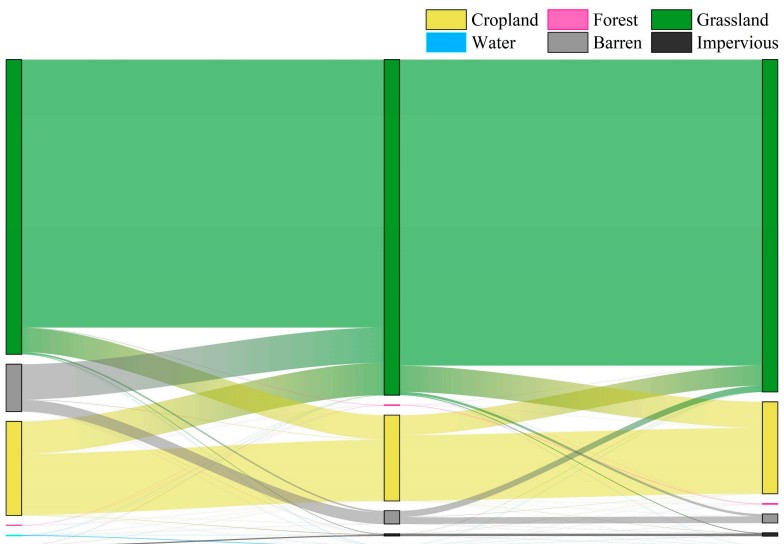

**Figure 13.** Land cover transfer mulberry map, 2000–2013–2022.

To explore the precise impact of land use transformation on vegetation cover, this study introduces the formula for the contribution rate of vegetation cover resulting from land use transfer (Equation (12)). It computes the contribution rate of vegetation cover from land use transformation in the Wuding River Basin during the periods 2000–2013 and 2013–2022 (Figure 14). Given that the predominant land types in the study area are grassland and arable land, the contribution rate of vegetation cover is relatively low due to the small area of other land types.

Figure 14a illustrates that the most substantial positive impact during the period 2000–2013 was the conversion of grassland to arable land, contributing 1.52 per cent. This was followed by the conversion of arable land to grassland and wasteland to grassland, contributing 1.29 per cent and 1 per cent, respectively. The conversions of grassland to woodland, water bodies, and wasteland to arable land all made positive contributions to vegetation cover. Conversely, the conversion of cropland to water bodies and impervious surfaces, along with the conversion of grassland to wasteland, had a negative impact on the vegetation cover. Notably, the most significant negative effect was observed in the conversion of cropland to impervious surfaces. It is worth mentioning that the mutual conversion of cropland and grassland both had positive impacts on vegetation cover. This may be attributed to the progress of fallow farmland and forest and grassland projects, leading to the conversion of grassland with low *FVC* to cropland in the study area [50], aligning with the *FVC* values of land use in this study.

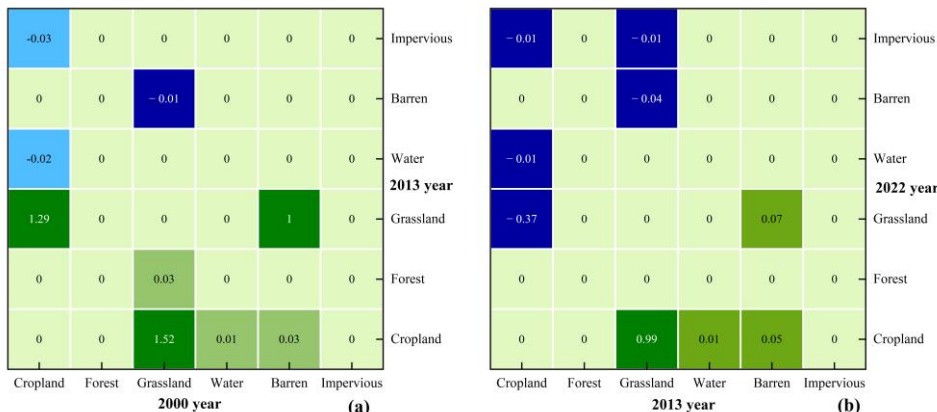

**Figure 14.** Vegetation cover contribution (data shown as a percentage). (**a**): Contribution of land type change to *FVC* in the 2000–2013 period; (**b**): Contribution of land type change to *FVC* in the 2013–2022 period. note: Each number represents the contribution to *FVC* from the conversion of the site type on the horizontal axis to the site type on the vertical axis, with positive values representing positive impacts and negative values representing negative impacts.

Figure 14b reveals that during the 2013–2022 period, the most substantial positive impact continued to be the conversion of grassland to cropland, contributing 0.99%, followed by the conversion of wasteland to grassland, cropland, and water bodies to cropland. The most significant negative impact was the conversion of cropland to grassland, contributing −0.37%, followed by the conversion of cropland to water bodies, impervious surfaces, and the conversion of grassland to wasteland and impervious surfaces. Overall, an increase in the area share of cropland types is more conducive to the expansion of vegetation cover in the study area, aligning with the experimental results of the risk detector.

## 4. Discussion

### 4.1. Effect of Natural Factors on FVC in the Wuding River Basin

The Loess Plateau region experiences an arid and semi-arid climate, and the scarcity of water resources constrains the robust growth of vegetation in arid areas [51]. The Wuding River Basin receives 300–500 mm of annual precipitation, increasing from northwest to southeast, corresponding with the spatial distribution of the *FVC*. Precipitation promotes vegetation growth by providing water and soil nutrients, leading to a transition from grassland to desert vegetation from south to north. However, uneven annual precipitation, with 65% occurring in July-September, limits normal vegetation growth. Sunshine hours also explain the spatial variability of the *FVC*, as sunlight directly affects plant photosynthesis. Adequate light enhances plant photosynthetic efficiency and boosts their growth rate. However, excessive sunshine hours can increase water evaporation rates, negatively impacting water availability to plants [52]. Additionally, excessively high sunshine hours may elevate surface temperatures, posing a threat to vegetation survival. Nevertheless, optimal temperatures can foster vegetation growth, promoting not only the photosynthetic process in conjunction with sunlight but also accelerating water utilization by plants. These meteorological factors collectively enhance the robust growth of vegetation and the stability of regional ecosystems [53].

In addition to meteorological factors, surface characteristics like soil and altitude influence vegetation cover. Different soil textures result in varied vegetation growth conditions due to differences in soil properties, affecting the region's vegetation cover. The study area features mylonitic and sandy soils, with mylonitic soils being the most widespread and serving as the primary cultivated soils [54]. Conversely, sheep soil type soils are loose and nutrient-poor, and windy sandy soils exhibit poor structure and low agricultural productivity, thereby limiting improvements in vegetation cover to some extent. Elevation stands out as a key topographic factor, influencing vegetation cover through its impact on factors like light, temperature, and precipitation. In this study, the vegetation

cover displayed a pattern of a gradual increase followed by a significant decrease with rising altitude, underscoring the notable influence of altitude on vegetation cover.

### 4.2. Effect of Human Factors on FVC in the Wuding River Basin

Both natural factors and human activities significantly influence vegetation cover. In the Wuding River Basin, the land use type factor, a human activity, exerts the strongest influence on vegetation cover, aligning with Zhao et al.'s findings [55]. Studies have indicated that the National Return of Ploughland to Forests and Grasslands Project, initiated in 1999, effectively restored grasslands and woodlands in northern Shaanxi, leading to a substantial improvement in regional vegetation cover [47]. Local government policies, such as forest closure and terrace construction, further enhanced vegetation cover in northern Shaanxi [56]. This underscores the crucial role of anthropogenic interventions, including returning farmland to forests and grasslands and soil and water conservation projects, in promoting vegetation growth and ecosystem health in northern Shaanxi. However, human activities, mainly land use, have a dual impact on regional vegetation cover, with both positive and negative outcomes. For instance, the Daliuta coal mine in the northern part of Yulin City, Shaanxi Province, initially caused environmental pollution, soil erosion and other geological hazards. Still, reclamation and remediation efforts since the 1990s have mitigated environmental degradation, resulting in the establishment of artificial woodland, scrub, and grassland [57]. In the long term, the conversion of land use types due to the project of returning farmland to forest and grassland and urban expansion will bring about varying degrees of interference and changes to the regional ecological environment. Effectively harnessing human initiative, managing the balance between resource development and ecological protection, is a necessary prerequisite for preserving vegetation cover.

### 4.3. Optimal Conditions for Vegetation Growth in the Wuding River Basin

We identified the best conditions for vegetation growth based on risk detection (Table 10). The factor interval with the largest average *FVC* was considered the optimal condition for vegetation growth [58]. The analysis reveals that the vegetation of the Wuding River Basin is suitable for growing in environments with moderate temperatures and sunshine, high rainfall and humidness. The maximum *FVC* was achieved when the topographic factors of the study area were irrigation-silting soil, elevation in the range of 850–990 m, and slope at about 20°. The vegetation in the study area is suitable for a low population density and low GDP environmental conditions, and cropland is the best land cover type for vegetation growth.

**Table 10.** Optimal factor intervals.

| Driven Factor | Suitable Type or Range of *FVC* | *FVC* |
|---|---|---|
| Tem | 18.0–20.0 °C | 0.486 |
| LST | 19.5–20.3 °C | 0.427 |
| Pre | 601–648 mm | 0.470 |
| RH | 53.4%–54.4% | 0.421 |
| SD | 7.86–8.07 h | 0.490 |
| Aridity index | 58%–66% | 0.441 |
| Soil | irrigation-silting soil | 0.647 |
| Soil erosion | 1.5%–5.1% | 0.416 |
| Elevation | 850–990 m | 0.491 |
| Slope | 18.6–22.8° | 0.412 |
| Pop | 32–59 persons/km$^2$ | 0.429 |
| LC | cropland | 0.575 |
| GDP | 0.93–2.58 | 0.391 |

*4.4. Limitations and Prospects*

Based on Landsat image data from the GEE platform, we analyzed and predicted spatial and temporal changes in the *FVC* in the Wuding River Basin, and explored the drivers of *FVC* by the natural environment and anthropogenic factors. We also propose a vegetation contribution formula to quantify the role of land use transformation on the regional *FVC*. While this study employed high-resolution image data and a precise experimental model, potential errors may arise from factors such as the external azimuthal displacement of the sensor, the curvature of the earth, and complex terrain [48]. Although we included some anthropogenic factors in the analysis and quantified the contribution of land transformation to vegetation cover, the analysis of the drivers of *FVC* change in the results was incomplete due to the diversity of anthropogenic activities, and further selection of more relevant anthropogenic factor indicators, the improvement of the model and analytical methods, and an exploration of the accuracy of the formulae for the contribution of vegetation cover are awaited in subsequent studies.

## 5. Conclusions

Based on the dimidiate pixel model, this paper explores the spatial and temporal pattern of vegetation cover and its change characteristics in the Wuding River Basin from 2000 to 2022, and at the same time, adopts the Geodetector method to explore the drivers that cause changes in vegetation cover. To study the effects of land use-dominated human activities on vegetation cover, we proposed an equation to quantify the contribution of land use transformation. Our results show the following:

[1] The overall trend of the *FVC* in the Wuding River Basin slowly increased from 2000 to 2022 (S = 0.0046, *p* < 0.01). The *FVC* increased rapidly until 2013 (S = 0.011, *p* < 0.01), then decreased slowly after 2013 (S = −0.0027, *p* < 0.01). The downstream area had the fastest growth and decline rates in both periods. There was gradual conversion of low-grade *FVC* to higher grades and an overall improvement in vegetation cover in the study area.

[2] From 2000 to 2022, the spatial distribution of the *FVC* in the Wuding River Basin showed a gradual increase from northwest to southeast, with obvious geographical differentiation. The high *FVC* areas were mainly distributed in the southeastern part, and the low areas were mainly distributed in the northern part of the study area. The future trend of the *FVC* is mainly decreasing (58.0%), which is mainly distributed in the downstream area of the Wuding River, such as Zizhou County and Mili County, where the vegetation cover is at a high risk of degradation, and needs to be strengthened for protection in the future.

[3] The primary factor influencing the spatial differentiation of the *FVC* in the Wuding River Basin from 2000 to 2022 was the land cover, which was directly affected by human activities showing an explanatory power of 0.345. Additionally, natural factors, including precipitation, soil, and sunshine, also contributed to some extent. The interactions among most of the driving factors exhibited either two-factor enhancement or non-linear enhancement, with the most robust interaction explanatory power observed between land use and precipitation, reaching an explanatory power of 0.523.

[4] The land use types in the study area, including grassland, water, forest, and impervious, exhibited an increasing trend in their area shares, while cropland and barren showed a declining trend. Grassland is growing the fastest and cropland is declining the fastest. The transformation of grassland into cropland made the most positive significant contribution to vegetation cover, at 1.52%. The expansion of cropland was more favorable for enhancing vegetation cover in the study area.

**Author Contributions:** Project administration and funding acquisition, J.X. and G.W.; writing—review and editing, J.X. and G.W.; writing—original draft and methodology, H.Z. and Z.H.; investigation, W.M. and Y.C. All authors have read and agreed to the published version of the manuscript.

**Funding:** This study was supported by the National Natural Science Foundation of China (grant number U21A2014); the National Natural Science Foundation of China (grant number 42271463).

**Data Availability Statement:** The data provided in this study are available on request from the corresponding author. The data are not publicly available as they are being collated.

**Conflicts of Interest:** The authors declare no conflicts of interest.

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
