# Peer review of "Analysis of Spatial and Temporal Changes in Vegetation Cover and Driving Forces in the Wuding River Basin, Loess Plateau"

_forests, doi:10.3390/f15010082_

Round 1

Reviewer 1 Report

Comments and Suggestions for Authors

The paper provides a study on vegetation change due to anthropogenic and natural reasons. It investigates spatial and temporal analysis in the Wuding River Basin in China. The paper provides a proper methodology, however, some missing parts need to be added. The authors should consider the suggestions below and answer the questions to improve the paper;

1.       The abstract should include the data information and the methods used for the purpose.

2.       Abbreviations should be given in their long form when first used. In lines 39-40 write the word for the letter F.

3.       Line 42: The sentence is not clear, the word amidst does not seem proper, please check the sentence.

4.       Page 2 Line 85: It is not clear what a geodetector is. Is this software or a model? Information should be given in the text and the methodology section.

5.       Page 2 Lines 86-89: the words “for the contribution of land use transfer to vegetation cover” are repeated. It can be removed.

6.       Page 4 line 123: How the NDVI data were inverted? Mention the satellite data.

7.       Page 6: The role of volatility should be given in the manuscript.

8.       Page 8 line 212: Fig. 2 should be Fig. 3.

9.       Table 1 has different data with different resolutions. How are they harmonized? It should be mentioned in the manuscript.

10.   There are two temperature factors in Table 5, what are their differences? Is the first one the temperature of the air?

11.   Table 6: What are Roll-in, Roll-out and Variation? How they are calculated is not clear, mention them in the manuscript.

12.   Page 10 line 278: Check the number of Figures.

13.   Page 11 line 291: It is not clear what the average FVC is divided into, please describe it.

14.   Page 12 line 339: The number of figure is missing.

15.   Figure 9: a) there are two strips in the figure, it looks like a systematic error. 9b) How the future trends are calculated is not given in the manuscript.

16.   Page 13 line 376: What is the q value used for and how is it obtained? It should be mentioned in the manuscript.

17.   Table 8: What do the letters Y and N in the table mean? They should be stated in the title of the table.

18.   Figure 11 is not cited in the manuscript.

19.   Figures 3 and 12: the quality of the figure is low, it should be improved.

20.   The reference list is not included in the end of the manuscript.

Reviewer 2 Report

Comments and Suggestions for Authors

General Comments:

The paper titled "Analysis of Spatial and Temporal Changes in Vegetation Cover and Driving Forces in the Wuding River Basin, Loess Plateau" provides valuable insights into the dynamics of vegetation cover in a crucial region, contributing to ecological protection in the Yellow River Basin. The study employs the dimidiate pixel model to assess changes from 2000 to 2022 and predicts future trends. Geodetector and a novel vegetation cover contribution formula are utilized to explore driving forces, particularly focusing on the impact of land transformation on vegetation cover. Overall, the paper is well-structured and presents significant findings; however, there are areas that require minor revisions.

Specific Comments:

Problem Description:

While the paper successfully outlines the importance of vegetation cover in the Loess Plateau region and its relevance to ecological protection, the problem description could be expanded to provide a more comprehensive context. What is the main problem of the study and how did you overcome?

Introduction Section:

The introduction section needs to incorporate more references to provide a solid theoretical foundation for the study. Include relevant literature on vegetation cover dynamics, dimidiate pixel modeling, and geodetector applications in similar ecological studies. This will strengthen the theoretical framework and demonstrate a thorough understanding of the existing body of knowledge.

Conclusion:

The paper makes a significant contribution to our understanding of vegetation cover dynamics in the Wuding River Basin. Addressing the suggestions for minor revisions will further strengthen the paper and ensure its alignment with the highest standards of academic rigor and clarity.

Reviewer 3 Report

Comments and Suggestions for Authors This paper used comprehensive data and methods to investigate the spatial and temporal changes of FVC, and its drivers. The topic is interesting and the results contain a wealth of information. Generally, I think the introduction section should provide more background information and clearly state the research gap. In the method section, some details are needed, as mentioned in the comments. The results can be more concise and focused.

11.Figure 2: Flowchart of technical route. There should be a clear connection between each part in Figure 2.

22.Line 339: Missing figure number.

33.The text and labels in most of the figure are not clear enough.

44.Table 7: The abbreviation should be explained.

55.The discussion (for example, Section 4.1) should be more related to the results of the study and what we can further learn from your results. The current text seems unrelated to your results, specifically, the sunshine hours did not show high explanatory power in the geodetector analysis.

66.LUCC reflects land use and cover change, so it cannot be a land cover type (Section 4.3).

77.It is hard to understand why the authors analyzed land-use type shifts independently; I mean, the factor could also be included in the geodector analysis. Or does it provide more information or different information for quantifying the drivers?

88.One of the limitations might be the different spatial resolutions of the dataset, even if some of them have the same resolution, the initial data source might have a very different spatial resolution.

99.I am very confused about what you want to answer: vegetation cover or vegetation cover change? I did not find a clear definition in your paper. What is your dependent factor in Geodetector? It should be noted that FVC and FVC change are very different, and the drivers are also very different. It should be clearly defined in your study.

110.What is the spatial resolution of your data? Do they have the same spatial resolution in Geodetector analysis?

Round 2

Reviewer 1 Report

Comments and Suggestions for Authors

The authors answered the questions and considered the suggestions. It is improved, but some parts still need clarification. The authors should check the comments below;

1.     It is mentioned that the role of volatility is given in the manuscript, however, I could not find it. Please write the response into the manuscript. Write the page, line information in your response.

2.     Line 49: the word “analysedanalyzed” should be corrected.

3.     In the manuscript, the word “Geodetector” should be checked. In some sentences it is used as a method, then it should be written as “Geodetector method”. In other sentences it is used as software, then it should be written as “Geodetector software. Section 2.2.5 mentions as a method, but in reference [12] (which is Chinese and I do not understand) it appears as software. Lines 85, 88, 114 are just some of them. Please check the manuscript.

4.     Table 1: I suppose they are resampled to 1 km. Is that right? Mention that in the title.

5.     Line 325: The word “divide” does not appear suitable. Do you mean “classified”? Please check it.

6.     Figures 3 and 12 still appear blurred. Please check the PDF version of the paper.

7.     It is mentioned that “q is the explanatory power of each factor in the geodetector's factor detection for the spatial distribution of FVC” please also add what it means when it is high or low in the manuscript.

8.     The format of the references should be checked, there are different types.

9.     The references are mostly China-oriented, and they should be updated for international readers. Please check similar international studies.

Reviewer 3 Report

Comments and Suggestions for Authors

1.     The authors stated that “The second is the exploration of the driving factors that caused the change of vegetation cover in the study area.”. So, what’s your dependent variable in Geodetector: the change trend of FVC (Theil-Sen Median) or average FVC of 2000-2022 or one year’s FVC? And what about the independent variables? Are they the change trends or average of 2000-2022 or one year’s data? I am sorry I did not find where you introduced these critical information. Therefore, I cannot clearly understand your drive factors analysis part.

2.     Do the spatial resolution of all data were resampled into 1km? It should be clearly stated.

3.       Line 230, there was no equation 14. In addition, the study have calculated Theil-Sen slope and change trend, I do not think it’s reasonable to use equation 10 to quantify the contributions of the land cover change, because the author have stated that the change of FVC has obvious turning points.
